# Microbiological Hazards in Dry Dog Chews and Feeds

**DOI:** 10.3390/ani11030631

**Published:** 2021-02-27

**Authors:** Jagoda Kępińska-Pacelik, Wioletta Biel

**Affiliations:** Department of Monogastric Animal Sciences, Division of Animal Nutrition and Food, West Pomeranian University of Technology in Szczecin, Klemensa Janickiego 29, 71-270 Szczecin, Poland; jagoda.kepinskapacelik@gmail.com

**Keywords:** pet foods, dog chews, microbiological hazards, safety, mycotoxins, bacteria, RASFF notifications, European Union

## Abstract

**Simple Summary:**

Food-borne infectious agents can affect the health of pets. Good quality pet foods have high nutritional value, including microbiological quality. The most important biological hazards in feed are *Salmonella, Enterobacteriaceae, pathogenic Escherichia coli, Staphylococcus aureus, Listeria monocytogenes, Clostridium perfringens, Clostridium botulinum, Aeromonas*, and *Campylobacter*. Other biological hazards of feed are the mycotoxin-producing molds. Although in some studies the level of mycotoxins in the foods does not exceed the maximum allowable amounts, long-term ingestion of mycotoxins may have adverse health effects, as these compounds accumulate in the tissues of animals. Animal by-products or derived products can also be a source of the above-mentioned risks. Pet foods products can also be a source of human infection. The close coexistence of humans and animals means that ensuring the safety of pet foods should be a priority.

**Abstract:**

Nowadays, dogs are usually equally treated with other family members. Due to the growing caregivers’ awareness, the pet foods industry is changing dynamically. Pet foods are manufactured with a myriad of ingredients. Few authors of scientific papers deal with the topic of foods products’ safety for pet animals, assessed from the perspective of their caregivers. Despite the many methods of producing foods of the highest quality, there are still cases of contamination of pet foods and treats. In the case of dried chews for dogs, bacteria of the genus *Salmonella* are the most common risk. In the case of both dry and wet foods, in addition to many species of bacteria, we often deal with mold fungi and their metabolites, mycotoxins. This article presents selected microbiological risks in dog foods and treats, and analyzes the Rapid Alert System for Food and Feed (RASFF) system (2017–2020) for pathogenic microorganisms in dried dog chews, treats and foods. In this period, pet food-related notifications were registered, which were categorized into different types. Analyzing the RASFF notifications over the period, it has been shown that there are still cases of bacterial contamination of dog foods and treats, while in terms of the overall mycotoxin content, these products may appear safe.

## 1. Introduction

The companion animal population is increasing worldwide. According to statistics reported by the Fédération Cynologique Internationale (FCI), the approximate total number of dogs (pure-breed or not) around the world is about 147 million [1]. The nutrition of dogs mainly involves dry foods and their industry is still growing—the annual growth rate of the pet food industry (average value over the past 3 years) is 2.6% [2]. An increasing share of these impressive numbers of animal owners is becoming more aware of the nutritional needs of their animals. The constant demand from owners for better quality products means that the pet food sector is becoming particularly aware of providing nutritious food for animal health and welfare.

In the era of globalization, we deal with international trade and a constantly growing flow of goods. This allows animal diseases and zoonoses to spread around the world. Previously eradicated animal diseases risk reintroduction into the European Union (EU) as significant amounts of animal foods products from endemic countries are continuously imported into the EU, both legally and illegally [3,4,5].

In the EU, control over the safety of raw materials and food products is carried out by relevant authorities, including Rapid Alert System for Food and Feed (RASFF). As reported by RASFF [6], pet foods might be a significant source of many hazards associated with biological, physical or chemical agents in animal feed. These factors can cause illness or injury in pets in the absence of adequate production control. 

Treats and chews are a regular part of our dogs’ diet [7]. In addition to providing the basic portion of foods, caregivers often reach for it during training or when it is necessary to keep a pet busy for a while. It should be kept in mind that these types of products should only be a small part of a dog’s diet because excessive consumption of treats can lead to obesity and even nutritional imbalance. It should be mentioned that, according to the law, such products are classified as complementary products.

Dogs have a continual desire to bite and chew. If we do not provide pets with chews, they may damage home furnishings or elements in the garden or elsewhere. There is a trend in many countries to feed dogs and cats a raw meat-based diet and to include treats that are animal by-products [7,8]. This is mainly due to perceptions of the health benefits for pets and the belief that dried chews are of natural origin. Important aspects are their specific smell, as pets love aromas as well as taste and texture. This proves the taste of dried chews. However, are they really safe for pets?

Dog chews consist of animal by-products (ABPs) or derived products. ABPs are materials of animal origin that are not intended for human consumption or that humans do not consume [9]. Derivative products are products obtained from at least one treatment, transformation or processing step of ABPs.

Not only are chews produced from animal by-products, but also meat and bone meals, fish meals, blood meals, blood products, animal fats and pet foods.

Currently, there is a wide range of dried natural chews and treats in supermarkets. They are primarily from many species of animals—ears, the trachea, tendons, masseters and much more. Such chews are usually sold loose, “in pieces”, without the original packaging. We do not know often their use-by date and storage time. These factors increase the risk of contamination of the products with pathogenic microorganisms. Raw dog foods are more likely to be contaminated because they are not subjected to rigorous processing procedures such as heating and sterilization. There is also a risk of dry foods, which can also become contaminated with bacteria after heat treatment [10].

Multiple research reports have exposed pet food quality problems and their influence on human and animal health [11,12,13,14,15]. In recent years, notifications of pathogenic microorganisms (bacteria, fungi, and the toxins that they produced) constituted about 20% of all notifications in RASFF, showing the presence of *Salmonella, Listeria, Escherichia* and others [6,16]. The objective of this paper was to characterize the presence of biological contaminants in dry dog chews and feeds and analyze the period from 2017–2020 using notifications made to RASFF database. This article presents selected microbiological risks in dog foods and treats and analyzes the RASFF system for pathogenic microorganisms in dried dog chews, treats and foods.

## 2. Microbiological Hazards in Treats

According to the European Pet Food Industry Federation (FEDIAF) [17], the list of biological hazards in pet foods that are reasonably likely to cause illness or damage to animals in the absence of monitoring include *Enterobacteriaceae, pathogenic Escherichia coli, Salmonella, Staphylococcus aureus, Listeria monocytogenes, Clostridium perfringens, Clostridium botulinum, Aeromonas, Campylobacter*, molds and yeasts. In reference also to reports in the RASFF system, the pet foods market poses a risk of pathogenic microorganisms (bacteria, fungi, and the toxins that they produce). Currently, there are no strict regulations on maximum limits of bacterial and fungal contamination for pet foods. The microbiological requirements currently in force do not allow the presence of *Salmonella* in five samples of feed weighing 25 g, and limit the number of *Enterobacteriaceae* in feed materials of animal origin from 10 to 300 cfu/g in two out of five samples of a tested batch [18]. Moreover, European Commission (EC) Regulation on the general rules of feed hygiene [19] does not apply to retail pet foods. So far, no limit values for mycotoxins in pet foods have been established. However, in 2006 the European Commission (EC) issued a recommendation on permitted levels of mycotoxins in animal feed [20]. In the case of other threats, such as *Listeria* or *Clostridium* bacteria, there are no legal acts setting limits for the presence of these bacteria in food products for dogs.

The issues concerning biological hazards occurring in chews concern mainly bacteria of the genus *Salmonella*. *Salmonella* spp. are Gram-negative bacteria of the *Enterobacteriaceae* family that can colonize the digestive tract of most vertebrates [21]. Dog chews are usually products with a long shelf life, which means that if they are contaminated with *Salmonella*, they constitute a long-term exposure of humans and dogs to a given pathogen present in the entire batch of the product. *Salmonella* spp. are the main bacterial zoonotic agent found in the feces of animals [22], from where they end up in water, soil and food. This can lead to the development of zoonoses in humans [23,24,25]. The route of infection may be direct contact with a contaminated chew, contact with the dog’s saliva containing pathogenic microorganisms, or contact with bacteria present in the animal’s stool [26].

There are many examples in the literature of the transmission of animal diseases to humans. This problem was noticed more than 20 years ago. An epidemiological investigation showed that *Salmonella enterica*, present in dog chews produced from pig ears, was the cause of salmonellosis in humans [27]. Pathogens in chews can be resistant to many antibacterial substances [28]. An example of such a situation is the case of 2002, which also took place in Canada, where from five people with symptoms of food poisoning, a strain of *Salmonella enterica*, serotype Newport, resistant to inter alia, ceftazidime, cefoxitin, ampicillin and chloramphenicol, was isolated. An epidemiological investigation carried out at that time showed that the source of the infection was the American-made beef chews found in the home of one of the patients [29].

Another example of the transmission of *Salmonella* from animals to humans was shown in the studies by Cavallo et al. [30]. They interviewed United States patients suffering from salmonellosis and performed laboratory tests. A total of 43 sick people were identified, of whom 95% indicated that they were exposed to dogs and 69% had contact with treats. In the next part of the investigation, 88% of the test subjects reported that they had been in contact with X’s chicken jerky. Testing of these products allowed the isolation of *Salmonella* from several analyzed patient-supplied samples. The inspection at company X revealed many irregularities in the chews production process, including improper processing of the raw material, the lack of basic hygiene rules in production, and the use of packaging that allows bacteria to multiply.

The research by Yukawa et al. showed the situation of contamination of chews with *Salmonella* in Japan [21]. Over 300 chews samples were collected, including both domestic and imported products. Laboratory analyses allowed for the conclusion that the treats may be contaminated with *Salmonella*, including strains resistant to antibacterial substances. The incidence of bacteria was 2% and was higher in imported than domestic products. It has been suggested that a necessary undertaking should be the implementation of an appropriate program of testing pet chews and treats for the presence of *Salmonella*. It is essential to take decisive action to prevent contaminated products from entering the market, which in turn may lead to salmonellosis not only in dogs but also in humans.

Infection with *Salmonella* as a result of eating a contaminated treat is problematic because it usually does not cause clinical symptoms in adult healthy dogs. Occasionally, as a result of prolonged exposure, gastroenteritis may occur. The clinical manifestations reported in individuals affected by salmonellosis vary widely and include severe fever, anorexia, abdominal pain, diarrhea (often hemorrhagic) and vomiting. Individual cases describing infection include conjunctivitis, bone and bone marrow inflammation and meningitis, as well as pneumonia, although initially it is difficult to associate these symptoms with infection with *Salmonella* [31,32,33,34,35].

An example of a less common but equally dangerous microbial threat is bacteria of the genus *Clostridium* [36,37,38,39]. Infection with bacteria of the genus *Clostridium* is increasingly recognized as the cause of diarrhea in outpatients [40]. Foods in the USA, Canada and Europe, and meat products intended for consumption by pets may be a likely source of contamination. This raises questions about the transmission of this pathogen to humans via food through the consumption of contaminated products [41]. It has been shown that bacteria of the genus *Clostridium* can be responsible for the spoilage of vacuum-packed chilled beef, lamb and venison and cooked meat products, chilled dog rolls packed in an oxygen-impermeable plastic casing. In studies to analyze bacterial contamination of bully sticks, 4% were contaminated with *Clostridium difficile*, while one was found to be infected with methicillin-resistant *Staphylococcus aureus* (MRSA) and one with a tetracycline resistant *Escherichia coli* [42,43].

Another example is *Listeria monocytogenes*, which is one of the pathogens isolated from food. In humans, it causes a disease called listeriosis. Animals that consume the contaminated food can be colonized with *L. monocytogenes* without showing clinical signs, making animals a possible source of infection in the household. Research by Bilung et al. [44], however, did not confirm the presence of these bacteria in dry and wet foods and treats for dogs.

Pet treats can be potentially dangerous, especially for children, due to the growing trend to produce dog treats similar to toys. Castrica et al. [45] conducted a study to assess the purchasing habits of dogs’ caregivers and determine whether and in what form pet treats could be potentially dangerous to humans, especially children. Research has shown that dogs’ caregivers are often influenced by the shape of the treats when shopping. Most of the respondents admitted that the treats are attractive to their children, and the possibility of children’s contact is quite likely. Some respondents admitted that there were cases of involuntary consumption of snacks for animals, with fever and gastrointestinal symptoms. Microbiological tests showed that the delicacies were in good sanitary condition, except for one sample where the presence of *Listeria ivanovii* was confirmed. From this it can be concluded that, despite the relatively low risk of microbial contamination, there is still the possibility of negative consequences from consuming dog treats. This is especially true for children who are most exposed to pet snacks, attracted by a toy-like shape.

Animal by-products may only be used when the associated risks to human and animal health have been minimized during processing prior to placement on the market. If this option is not available, ABPs should be disposed of under safe conditions [9]. This means that if dangerous levels are detected in each batch of product, e.g., of the genus *Salmonella*, this batch of product cannot be placed on the market. Exposure of both dogs and humans has been shown to increase significantly when bacterial contamination occurs after or during coating the treats with fat [46].

Animal by-products intended for dog treats must be sufficiently heat-treated to minimize the risk of survival of pathogenic microorganisms. Producers also need to prevent contamination after heat treatment [47]. Commercial documents and certificates accompanying ABPs and derived products during transport must contain at least information on the origin, destination and quantity of such products, and their description [9]. Action is needed to address emerging public health problems related to pet foods and the effectiveness of mitigation measures. The results of such models can form the basis for improving production processes, informing consumers about risks and regulatory actions [48].

Not only dogs are exposed to foods contaminated with pathogenic microorganisms [49,50]. Research confirms the disturbing presence of microbiological hazards in compound feed [51]. The analyses carried out by a team of scientists were aimed at assessing the microbiological quality of feed mixtures used in Poland in 2007–2010. The presence of *Salmonella* sp. and the number of *Enterobacteriaceae* bacteria as well as mold fungi were estimated. The percentage of contamination of feed mixtures for poultry, pigs and cattle by *Salmonella* sp. ranged from 0% to 3.5%. The highest level of *Enterobacteriaceae* contamination was found in wet pet foods. These analyses allowed for the conclusion that the microbiological quality of feed mixes is getting better and better compared to previous years.

## 3. Pet Foods—Contamination with Mycotoxins

It should be mentioned that not only chews can be a source of microbiological threats to dogs. Maintenance foods can also be dangerous, not only in terms of the presence of bacteria. More and more research is being conducted on the presence of mycotoxins in the foods of domestic animals. As reported by Pigłowski, mycotoxins were the most frequently reported hazard category in RASFF in 1981–2017. The largest number of notifications was related to aflatoxin B1 [52].

Mycotoxins are harmful metabolites of mold fungi that are found in various foods. Most mycotoxins belong to three types of fungi—*Aspergillus, Penicillium*, and *Fusarium*. The most disturbing substances are aflatoxins, vomotoxins, ochratoxins, zearaleon, fumonisin [53]. The large group of EU recommendations on safe levels in animal products only apply to three mycotoxins (Table 1).

Mycotoxins are produced in cereal grains, as well as in fodder before, during and after harvesting in various environmental conditions [55]. In the case of cereals, it has been shown that most of the impurities are close to the surface of the grains. By removing only part of the outer layers of the grains, microbial contamination can be significantly reduced [56]. Research by Oliveira et al. [57] showed that metabolites of lactic acid bacteria (LAB) are a reliable alternative for reducing fungal infections before and after harvest.

The presence of mycotoxins in feed may reduce feed consumption and adversely affect the health of the animals [58,59,60]. Their presence can also cause inhibition of the total weight gain [61]. In addition, the possible presence of toxic residues in edible animal products such as milk, meat and eggs can have a detrimental effect on human health [62,63]. Contamination with fungi and their metabolites in the form of mycotoxins affects both the organoleptic properties and the nutritional value of the feed and carries the risk of poisoning. Studies have shown that a high percentage of feed samples are contaminated with more than one mycotoxin [64]. The effects of consumption of foods contaminated with mycotoxins depend on the amount of toxins present and the duration of exposure, as well as the individual sensitivity of the animals. Mycotoxins have a variety of chemical structures responsible for their various biological effects. Depending on their exact nature, these toxins can be carcinogenic, teratogenic, mutagenic, immunosuppressive, trembling, hemorrhagic, hepatotoxic, nephrotoxic and neurotoxic [65,66]. For example, aflatoxin B1 has a strong hepatotoxic effect, however, it has been shown that it can be minimized by supplementing the feed with curcumin [67]. Controlling the growth of mold fungi and the production of mycotoxins is very important for the feed producer and for the animals [68]. The study by Singh and Chuturgoon [69] aimed to compare the microbiological quality of standard feeds with premium feeds. These studies showed that regardless of the brand, all foods samples were contaminated with fungi (mainly *Aspergillus flavus*, *Aspergillus fumigatus* and *Aspergillus parasiticus*) and mycotoxins (most often aflatoxins and fumonisins). The obtained results suggest that more expensive dog foods do not provide the highest quality, nor do they guarantee microbiological purity. On the other hand, research by Leiva et al. [70] has shown that in the case of dry feed, the extrusion process may be helpful in reducing the pathogenicity of microorganisms and does not affect the digestibility of the feed.

In another studies, an alarming presence of fumonisins in animal feed was found [71]. The simultaneous contamination of the feed with fumonisin fractions was also quite common. It should be kept in mind that toxins show a synergistic effect, especially since the coexistence of fumonisin with other *Fusarium* sp. toxins is a real possibility.

Studies by Witaszak et al. [72] confirmed the presence of five types of fungi-producing mycotoxins in the amounts permitted by EU regulations. However, the low level of mycotoxins in dog foods does not eliminate the risk and caution should be exercised because long-term daily consumption of even small amounts of mycotoxins can lead to slow damage to the animal’s body and the development of many diseases, including cancer. The presence of mycotoxins in foods was confirmed by a study by Shao et al. [73]. Only one of 32 samples was free from mycotoxin contamination. Moreover, all other samples were contaminated with at least three different types of mycotoxins.

Analyses by Tegzes et al. [74] aimed to compare cereal and cereal-free dog foods in terms of their mycotoxin content. The test results confirmed the presence of mycotoxins in dry cereal dog foods, while in cereal-free foods they were not found. This study suggests that the risk of exposure to mycotoxins is higher with dry dog foods containing cereals. To minimize the risk, dog food manufacturers should choose grain types that are less susceptible to the presence of mycotoxins.

Cereals are usually the main component of veterinary foods, intended for dogs with various diseases that require safe and wholesome nutrition. Cereal grains can often be contaminated with *Fusarium* fungi, which can produce mycotoxins. In studies by Witaszak et al. [75], samples of veterinary feeds were examined for the presence of mold species and mycotoxins. Only 9.5% of the samples were free from mycotoxins produced by Fusarium, however, none of the tested samples exceeded the permissible limits of mycotoxin content in feed, as defined by EU regulations. This means that it is necessary to systematically test both domestic animal and veterinary feeds in terms of the content of harmful microorganisms and their metabolites, because especially veterinary feeds should be characterized by the highest level of safety for animals.

Studies by Macías-Montes et al. [76] showed that the presence of mycotoxins is quite common in dry dog foods. However, the concentrations of most of them are among the lowest reported so far. It was proven that the mycotoxin content was not influenced by the feed quality. Chronic exposure to mycotoxins and their hidden forms may prove problematic.

The results of Okuma et al. [77] research revealed a low incidence of aflatoxin and ochratoxin in commercial pet foods. Although deoxynivalenol has been detected in many trials, its levels were well below those that can cause acute toxic effects.

A study by Gazzotti et al. [78] showed that all samples of extruded complete dog foods were compliant with current European legislation on mycotoxin contamination. However, these results revealed the need for further research into the potential risks of chronic low-dose exposure to various types of mycotoxins to which pet species are currently exposed.

## 4. Analysis of the RASFF System

Nowadays, the necessary tasks of companies involved in the production of food not only for people but also for animals is to ensure the quality and safety of food [79]. In the European Union, as part of the implementation of the Regulation (EC) 178/2002 [80], the Rapid Alert System for Food and Feed (RASFF) has been established to support the control and safety of food and animal feed on the European market. The RASFF system was established by the European Commission to quickly inform Member States about the risks associated with products that do not meet safety requirements and pose a risk to the consumer [80]. RASFF is a system for the exchange of information between official control authorities in Europe, which are members of this system. Information about food, feed and food contact materials potentially hazardous to human, animal health or the environment is entered here, as well as the follow-up activities as a result of the identification of such products. In a situation where a risk related to food, feed or food contact material/material is detected, the national contact point of a given network member is to prepare the so-called notification on a notification form created especially for this purpose and forward it immediately to the contact point of the European Commission.

RASFF notifications are divided into [81]:–Alerts are sent when a food or feed presenting a serious health risk is on the market and when rapid action is required;–Information is sent when hazardous food or feeds or materials/articles met food are identified, but it is not necessary to take immediate action in this regard in another country being a member of the network, e.g., because a given product is no longer available on the market or is only on the market of the country notifying the notification; and–Border rejection means notification of rejection of a batch, container or food cargo.

The notifications listed may be original or complementary. The results of RASFF’s activities are published in the form of reports that provide information on the number of notifications in the countries of notification of the product and its countries of origin, and on the identification of the risk to human and animal health posed by this product.

In our study, 66 notifications of bacterial contamination in pet foods and treats [82] between 1 December 2017 and 24 December 2020 were analyzed (Appendix A). Of these, 46 were dog chews. Alert notifications were the most numerous, while the fewest reports concerned product rejection at the border (Figure 1).

In most cases, the reporting country was Austria and Germany. The remaining reports were from Belgium, Italy, Sweden, Netherlands and Poland. The other reports were from Croatia, Norway, Slovenia, Lithuania, Czech Republic and the United Kingdom. Most reports concerned dangerous levels of *Salmonella*, while rest of them included the identification of both *Salmonella* and other *Enterobacteriaceae* in the same batch. Worryingly, as much as 29% of the notifications of contaminated chews, treats and foods related to products originating in Poland. The other notified products were introduced to the market from Germany, Netherlands, Turkey, UK, India and Thailand, Spain and Belgium, Brazil, Italy, China, Mexico, Slovakia, Croatia (Figure 2).

In our study, RASFF submissions for mycotoxins from 1 December 2017 to 24 December 2020 were analyzed and none were for dog foods, chews, or treats.

Despite the many reported cases over the time period analyzed, studies suggest that the incidence of *Salmonella* in animal products is decreasing [83]. This may result from the growing awareness of both producers and distributors regarding the production and storage conditions for dried chews. In their production, modern technologies can be used, which “destroy” the bacteria present on the raw materials, and the storage conditions effectively prevent their survival and development.

The RASFF system is a useful threat monitoring tool. However, it should be kept in mind that it has both strengths and weaknesses. Caution is needed when using the RASFF database both as a forecasting tool and for trend analysis, as changes in food law affect the frequency of regulatory sampling related to border and national controls. The involvement of countries in the RASFF database is variable, which often generalizes the observed trends. An important aspect is also that importing countries are raising market standards, therefore the exporting countries themselves have wider ramifications for food safety [84].

There are significant differences in food safety practices between countries, including both the number and type of contributions to the RASFF database, some of which are relatively very active in the border notification class. These findings should underpin EU food safety enforcement policy and practices [85].

In view of the complexity of the trade flows, the weakness of the RASFF system is observed, which is due to the high percentage of incorrect labels, together with a low number of label mismatches reported by RASFF. It would therefore be advisable to improve the system by reorganizing the control plans at all levels of the commercial chain. This could be made more feasible with research aimed at identifying relevant goals, working towards a global standardization of procedures and conventions [86].

In summary, the RASFF system, despite its imperfections and weaknesses, is an important tool that enables immediate action to be taken in relation to the occurrence of a threat. It allows for the quick elimination of products hazardous to health from the area of the European Union and ensures a uniform level of food safety throughout the territory of the EU. Data from the RASFF system are analyzed and constitute the basis for making changes in the field of European Union food law.

## 5. Conclusions

Analyses of the RASFF system notifications (2017–2020) showed that there are still cases of bacterial contamination (*Salmonella* and *Enterobacteriaceae*) of both pet foods and treats, while in terms of the overall mycotoxin content, these products may appear safe. In order to minimize the risk to the health of pets, the priority should be prevention, i.e., systematic testing of the raw materials and feeds, in terms of the content of harmful microorganisms and their metabolites. One form of simple risk reduction is, for example, leak testing of treat packages. Caregivers of potential consumers may also use the RASFF system to obtain information on whether a given risk has been recently identified in a given country.

## Figures and Tables

**Figure 1 animals-11-00631-f001:**
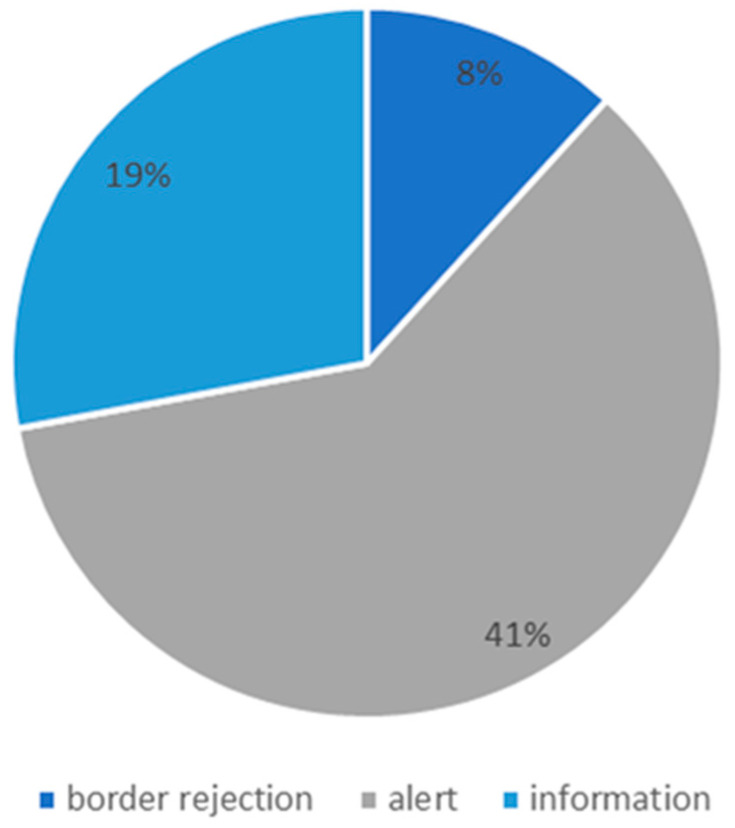
Pet foods and treats notifications of bacterial contamination in the Rapid Alert System for Food and Feed (RASFF) database (2017–2020).

**Figure 2 animals-11-00631-f002:**
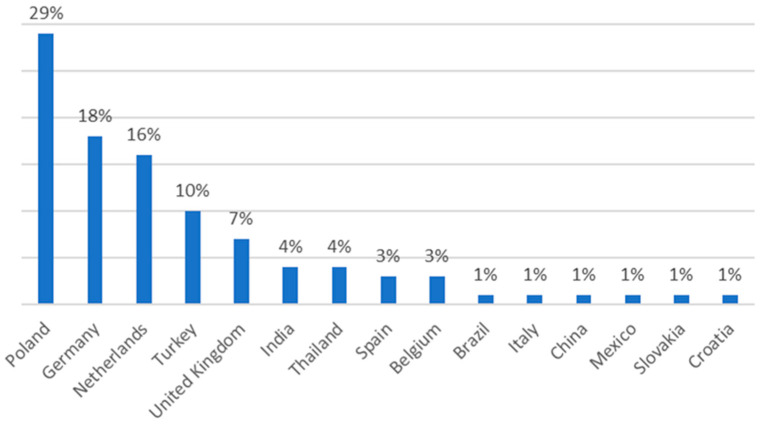
Countries concerned by the notifications of bacterial contamination in the RASFF database (2017–2020).

**Table 1 animals-11-00631-t001:** Guideline limit values for deoxynivalenol, zearalenone, ochratoxin A in pet products [54].

Mycotoxin	Pet Foods Product	Guide Value in mg/kg for a Feed with a Moisture Content of 12%
deoxynivalenol	cereals and cereal products with the exception of maize by-products	8
	maize-by products	12
	compound feed	5
zearalenone	cereals and cereal products with the exception of maize by-products	2
	maize-by products	3
	compound feed for adult dogs and cats other than those intended for reproduction	0.2
	compound feed for puppies, kittens, dogs and cats intended for reproduction	0.1
ochratoxin A	cereals and cereal products	0.25
	compound feed for dogs and cats	0.01

## Data Availability

Not applicable for writing this review article.

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
