# Peer review of "Microbiological Hazards in Dry Dog Chews and Feeds"

_animals, 2021, doi:10.3390/ani11030631_

Round 1

Reviewer 1 Report

Comments to authors:

In the present study, the authors evaluated and estimated “the exposure to microbiological hazards in dry dog chews and feeds”. Authors stated that the most important biological hazards in feed are Salmonella, Enterobacteriaceae, pathogenic Escherichia coli, Staphylococcus aureus, Listeria monocytogenes, Clostridium perfringens, Clostridium botulinum, Aeromonas, and Campylobacter. Besides, other biological hazards are mycotoxin-producing molds. The study is interesting and the information is useful.

I would suggest minor modifications to further improve the clarity of the manuscript.

-line no. 46, please replace “for” with “in” (“…injury in pets….”).

-Some of the references are not recent. Update it especially for mycotoxin (like 10.3389/fphar.2017.00143 and many other places), For example, line no-14. “The largest number of notifications was related to aflatoxin B1”. Insert a reference or revise/delete it.

-Authors are advised to check carefully the spelling/spaces and minor grammatical mistakes.  

Author Response

Dear Reviewer,

We would like to kindly thank you for the insightful review of our manuscript. Below we attached the list of changes made according to your suggestions. In the revised version of the manuscript we have marked the corrected parts of the text in the track change mode.

In behalf of co-author, once again thank you for your valuable efforts

Wioletta Biel

Reviewer:

Comments to the Author

Point 1: line no. 46, please replace “for” with “in” (“…injury in pets….”).

Response: We corrected this sentence.

Point 2: Some of the references are not recent. Update it especially for mycotoxin (like 10.3389/fphar.2017.00143 and many other places), For example, line no-14. “The largest number of notifications was related to aflatoxin B1”. Insert a reference or revise/delete it.

Response: We have updated the references;

[7]. Dodd, S.; Cave, N.; Abood, S.; Shoveller, A.K.; Adolphe, J.; Verbrugghe, A. An observational study of pet feeding practices and how these have changed between 2008 and 2018. Vet Rec. 2020, 186, 643-643, doi:10.1136/vr.105828.

[8]. Domesle, K.J.; Young, S.R.; Ge, B. Rapid screening for Salmonella in raw pet food by loop-mediated isothermal amplification. J. Food Prot. 2020, doi:10.4315/JFP-20-365.

[53]. Vudathala, D.; Klobut, J.; Cummings, M.; Tkachenko, A.; Reimschuessel, R.; Murphy, L. Collaborators, multilaboratory evaluation of a lateral flow method for aflatoxin B1 analysis in dry dog food. J. AOAC Int. 2020, 103(2), 480-488, doi:10.5740/jaoacint.19-0020.

[60]. Vudathala, D.; Cummings, M.; Tkachenko, A.; Guag, J.; Reimschuessel, R.; Murphy, L. A lateral flow method for aflatoxin B1 in dry dog food: an inter-laboratory trial. J. AOAC Int. 2021, qsaa175, doi:10.1093/jaoacint/qsaa175.

[61]. Barany, A.; Guilloto, M.; Cosano, J.; de Boevre, M.; Oliva, M.; de Saeger, S.; Fuentes, J.; Martínez-Rodriguez, G.; Mancera, J. Dietary aflatoxin B1 (AFB1) reduces growth performance, impacting growth axis, metabolism, and tissue integrity in juvenile gilthead sea bream (Sparus aurata). Aquaculture 2021, 533, 736189, doi:10.1016/j.aquaculture.2020.736189.

[64].Cheli F. Mycotoxin contamination management tools and efficient strategies in feed industry. Toxins 2020, 12(8), 480, doi:10.3390/toxins12080480.

[67].Muhammad, I.; Sun, X.; Wang, H.; Li, W.; Wang, X.; Cheng, P.; Li, S.; Zhang, X.; Hamid, S. Curcumin successfully inhibited the computationally identified CYP2A6 enzyme-mediated bioactivation of aflatoxin B1 in arbor acres broiler. Front. Pharmacol. 2017, 8, 143. doi:10.3389/fphar.2017.00143.

Point 3: Authors are advised to check carefully the spelling/spaces and minor grammatical mistakes.

Response: We have edited the manuscript according to your suggestion and corrected the grammatical mistakes.

Reviewer 2 Report

The issue of micro hazards in dry dog chews and feeds is certainly one worthy of consideration. However, I found the current paper confusing in its intent. If we are to treat this paper as a review, I would not expect to see RASFF data analysis, and it adds nothing to a review. And if the data is to be presented, then the paper is now a research paper and can not be accepted in its current form. To my mind, there are two:

  1. Keep the submission as a review but remove the RASFF data and make it clear in the title, abstract and simple summary this is a review not an estimation of exposure.
  2. Change to a research paper and present the RASFF data which would require that the 1) literature review be shortened considerably, 2) detail be provided on the RASFF database and how data was extracted and 3) more substantial analysis.

Some specific comments below for consideration:

  • The simple summary, abstract and introduction all serve different purposes, but, I would expect some continuity between the three, which is not the case. For example, the simple abstract starts with food-borne infectious agents affect dogs, while the abstract starts with pets being family members. In the introduction, there is mention of globalisation and trade which did not feature at all in either the abstract or simple summary.
  • The introduction includes several statements that need to have a reference or a more appropriate reference. For example, Line 47, how do you know treats and chews are a part of the diet. I don't believe the paper used as a reference does provide evidence of an increase in raw diet [Reference 5 Line 53-56] provides the evidence. Further, how do you know what people choose raw diets this is conjecture. There are other examples throughout the introduction.
  • If data from the RASFF is used, then the description needs to be included. For example, it not clear to me precisely what is reported as the text line 310 says 'when dangerous food or feed… comes in contact with food'.
  • How does food come in contact with food? In terms of analysis which notifications were used. More detail as to how the data was obtained is required what fields were available to the researchers.
  • In terms of analysis and presentation, the text mustn't replicate what is in figures and charts. Further, you should present confidence intervals for all percentages.
  • If data from RASFF is to be presented there needs to be a discussion of its strengths and weakness. Specifically, that notification systems are biased and often under-reporting and as such care is needed when interpreting the data.
  • The conclusion is presenting statements that should have been explored in the body of the papers. I agree that the quality of chews is determined by the raw material and storgage evidence, but I see nowhere in the paper a reasoned argument leading to this conclusion.

Author Response

Dear Reviewer,

Thank you for your valuable suggestions and comments for the improvement of the current manuscript. We have improved the manuscript based on your comments. All the mentioned changes have been incorporated in the manuscript. We appreciate your warm work earnestly and hope that the correction will meet with approval.

In behalf of co-author, once again thank you for your valuable efforts.

Yours sincerely,

Wioletta Biel

Response to Reviewer

Reviewer: The issue of micro hazards in dry dog chews and feeds is certainly one worthy of consideration. However, I found the current paper confusing in its intent. If we are to treat this paper as a review, I would not expect to see RASFF data analysis, and it adds nothing to a review. And if the data is to be presented, then the paper is now a research paper and cannot be accepted in its current form. To my mind, there are two:

  1. Keep the submission as a review but remove the RASFF data and make it clear in the title, abstract and simple summary this is a review not an estimation of exposure.
  2. Change to a research paper and present the RASFF data which would require that the 1) literature review be shortened considerably, 2) detail be provided on the RASFF database and how data was extracted and 3) more substantial analysis.

Response: Thank you for the suggestion. However, we decided to leave this paragraph with the RASFF system analysis, as well as the whole topic in the manuscript, since there was no such suggestion in other two reviews. Of course, if the Editor implies and the Reviewer upholds his decision, we will remove it.

Comments to the Author

Point 1: The simple summary, abstract and introduction all serve different purposes, but I would expect some continuity between the three, which is not the case. For example, the simple abstract starts with food-borne infectious agents affect dogs, while the abstract starts with pets being family members. In the introduction, there is mention of globalisation and trade which did not feature at all in either the abstract or simple summary.

Response: Thank you very much for your valuable attention. We corrected the introduction to maintain continuity.

Point 2: The introduction includes several statements that need to have a reference or a more appropriate reference. For example, Line 47, how do you know treats and chews are a part of the diet. I don't believe the paper used as a reference does provide evidence of an increase in raw diet [Reference 5 Line 53-56] provides the evidence. Further, how do you know what people choose raw diets this is conjecture. There are other examples throughout the introduction.

Response: We corrected for references that provided relevant evidence:

[7]. Dodd, S.; Cave, N.; Abood, S.; Shoveller, A.K.; Adolphe, J.; Verbrugghe, A. An observational study of pet feeding practices and how these have changed between 2008 and 2018. Vet Rec. 2020, 186, 643-643, doi:10.1136/vr.105828.

[8]. Domesle, K.J.; Young, S.R.; Ge, B. Rapid screening for Salmonella in raw pet food by loop-mediated isothermal amplification. J. Food Prot. 2020, doi:10.4315/JFP-20-365.

The publication of the authors Milanov, D.S.; Aleksić, N.R.; Vidaković, S.S.; Ljubojević, D.B.; Cabarkapa, I.S. Salmonella spp. in pet feed and risk it poses to humans. Food Feed Res. 2019, 46 (1), 137-145, has been removed.

Point 3: If data from the RASFF is used, then the description needs to be included. For example, it not clear to me precisely what is reported as the text line 310 says 'when dangerous food or feed… comes in contact with food'.

Response: Thank you very much. We corrected it [and please see our response to the Point 4]

Point 4: How does food come in contact with food? In terms of analysis which notifications were used. More detail as to how the data was obtained is required what fields were available to the researchers.

Response: We explained by correcting the previous point.

Point 5: In terms of analysis and presentation, the text mustn't replicate what is in figures and charts. Further, you should present confidence intervals for all percentages.

Response: We kindly thank you for this valuable comment. We removed duplicate content in the text.

We know that the confidence interval for the mean is a perfect supplement to the information about the mean, because it not only takes into account the average size of the mean from the sample to the mean from population, but also takes into account the standard deviation. However, in such a case, when notifications come from different countries, without creating a homogeneous experimental design, calculating the confidence interval in this case seems inappropriate. Since notifications obtained for individual countries do not have duplicates, it also becomes impossible to determine the standard deviation of individual data, and, consequently, to estimate the confidence interval.

Point 6: If data from RASFF is to be presented there needs to be a discussion of its strengths and weakness. Specifically, that notification systems are biased and often under-reporting and as such care is needed when interpreting the data.

Response: Very important point. The mentioned issue has been considered.

Point 7: The conclusion is presenting statements that should have been explored in the body of the papers. I agree that the quality of chews is determined by the raw material and storage evidence, but I see nowhere in the paper a reasoned argument leading to this conclusion.

Response: Thank you for this comment. The text has been corrected.

We have improved the manuscript based on your suggestions. All the mentioned changes have been included in the manuscript. We appreciate your work and hope that corrections will meet with approval. We kindly thank you for all valuable comments.

Reviewer 3 Report

This is an important review and reminder paper about the risks of microbial contamination of foods and feed. The paper focuses on pet foods, treats and chews, primarily for dogs, but also discusses other species including human exposure risks.

General Comments There is significant redundancy in the text and several points are restated or alluded to throughout. The text could thus be shortened by about a third to focus mostly on the results of the authors' summary of RASFF and other current findings. Secondly, while the authors' emphasis is on dogs here, most of the references and text actually refer to both dogs and cats. Except where the reference specifically refers only to dogs, like #66, please change "dogs" in the text to "dogs and cats" or "pets".

Some language and grammatical phrasing should be reworked or clarified.  The reference list is comprehensive and informative.

Specific Comments  Rework language where sentences or meaning may be unclear or incomplete -- e.g. Line 11. " means it has high -- ". " Line 17-18. Incomplete sentence-- you could remove the word "which". Line 18 . Remove 
"dogs" as it applies to both dogs and cats, and even other pets.

Throughout the text, the preferred term for pet "owners" is now "caregivers" or "guardians". 

The word "food" should be "foods " in most places. Please use italics for the names of all microbial species listed.

Please provide the full wording for abbreviations like FEDIAF and EC , when first used , followed by the designated abbreviation.  While the abbreviation, ADPs, is stated on Line 60, it is often not used in many places thereafter.

Please delete the wording "your " (Line 50).

Line 51. Suggest "obesity and even nutritional imbalance". Line 53. "dogs have a continual desire to bite and chew". Line 54, "-- elements in the garden or elsewhere." Lines 57-58. Suggest "Important aspects are their specific smell, as pets love aromas as well as taste and texture." Line 113. Suggest "This problem was noticed more than 20 years ago." Line 138. "-- pet chews and treats , -- ".

Line 208.  3. Pet foods ----" 

Line 222. "Mycotoxins are produced --- ". Line 274. "-- both domestic animal and veterinary -- ".

Lines 282-284 and 290-292 are redundant comments. Line 361. Suggest "--date, as this will provide beneficial information for both ourselves and pets." 

Author Response

Dear Reviewer,

We would like to kindly thank you for the insightful review of our manuscript. Below we attached the list of changes made according to your suggestions. In the revised version of the manuscript we have marked the corrected parts of the text in the track change mode.

Yours sincerely,

Wioletta Biel and co-author

Reviewer:

Comments to the Author

Point 1: There is significant redundancy in the text and several points are restated or alluded to throughout. The text could thus be shortened by about a third to focus mostly on the results of the authors' summary of RASFF and other current findings. Secondly, while the authors' emphasis is on dogs here, most of the references and text actually refer to both dogs and cats. Except where the reference specifically refers only to dogs, like #66, please change "dogs" in the text to "dogs and cats" or "pets".

Response: Thank you for your valuable attention. The authors carefully reviewed the text and references and revised the references that apply to both dogs and cats or all animals.

Point 2: Some language and grammatical phrasing should be reworked or clarified.

Response: The authors took this into account (+ see next responses).

Point 3: Rework language where sentences or meaning may be unclear or incomplete -- e.g. Line 11. " means it has high -- ".

Response: We corrected it.

Point 4: " Line 17-18. Incomplete sentence-- you could remove the word "which".

Response: We corrected it.

Point 5: Line 18. Remove "dogs" as it applies to both dogs and cats, and even other pets.

Response: We corrected it

Point 6: Throughout the text, the preferred term for pet "owners" is now "caregivers" or "guardians".

Response: Thank you for your valuable attention. We changed the word “owners” to “caregivers” throughout the text.

Point 7: The word "food" should be "foods " in most places.

Response: We changed the word “food” to “foods”.

Point 8: Please use italics for the names of all microbial species listed.

Response: We corrected it.

Point 9: Please provide the full wording for abbreviations like FEDIAF and EC, when first used, followed by the designated abbreviation.  While the abbreviation, ABPs, is stated on Line 60, it is often not used in many places thereafter.

Response: We provided the full wording for “FEDIAF” and “EC”. We used the abbreviation “ABPs” in some places.

Point 10: Please delete the wording "your " (Line 50).

Response: We deleted the word.

Point 11: Line 51. Suggest "obesity and even nutritional imbalance".

Response: We corrected it.

Point 12: Line 53. "dogs have a continual desire to bite and chew".

Response: We corrected it.

Point 13: Line 54, "-- elements in the garden or elsewhere."

Response: We corrected it.

Point 14: Lines 57-58. Suggest "Important aspects are their specific smell, as pets love aromas as well as taste and texture."

Response: We accepted the suggestion.

Point 15: Line 113. Suggest "This problem was noticed more than 20 years ago.

Response: We corrected it.

Point 16: " Line 138. "-- pet chews and treats, -- ".

Response: We accepted the suggestion.

Point 17: Line 208.  3. Pet foods ----"

Response: We changed the word “dog” to “pet”.

Point 18: Line 222. "Mycotoxins are produced --- ".

Response: We corrected it.

Point 19: Line 274. "-- both domestic animal and veterinary -- ".

Response: We accepted the suggestion.

Point 20: Lines 282-284 and 290-292 are redundant comments.

Response: We corrected it.

Point 21: Line 361. Suggest "--date, as this will provide beneficial information for both ourselves and pets."

Response: We corrected it.

Round 2

Reviewer 2 Report

I can see the authors have made a number of changes. However, the major issues have not been addressed and that that is the paper is by no means an estimate of exposure. It is a review of literature and a basic analysis of notifications of contamination. The title needs to better reflect the actual paper. Secondly, remains my view that a description of the notifications data is not appropriate in a review article.  The authors disagree with me which is reasonable. They noted that that the other reviewers did not make this point. They also indicated that the issue needed the editor to advise something I support. There is no value in forcing authors and reviewers to back and forward on what is clearly an editors decision.  

Author Response

Dear Reviewer,

We would like to kindly thank you for the insightful review of our manuscript. All the mentioned changes have been incorporated in the manuscript. We appreciate your warm work earnestly and hope that the correction will meet with approval.

In behalf of co-author, once again thank you for your valuable efforts

Wioletta Biel

Response to Reviewer 

Reviewer: I can see the authors have made a number of changes. However, the major issues have not been addressed and that that is the paper is by no means an estimate of exposure. It is a review of literature and a basic analysis of notifications of contamination. The title needs to better reflect the actual paper. Secondly, remains my view that a description of the notifications data is not appropriate in a review article.  The authors disagree with me which is reasonable. They noted that that the other reviewers did not make this point. They also indicated that the issue needed the editor to advise something I support. There is no value in forcing authors and reviewers to back and forward on what is clearly an editors decision

Response: Thank you for the suggestion. We corrected the title from “Estimation of exposure to microbiological hazards in dry dog chews and feeds“ to “Microbiological hazards in dry dog chews and feeds“. We also added some information about the RASFF system weaknesses. We decided to leave this paragraph with the RASFF system analysis, as well as the whole topic in the manuscript, since there was no such suggestion in other two reviews. The Editor also did not mention about removing this part of our manuscript.

We appreciate your work and hope that corrections will meet with approval. We kindly thank you for all valuable comments.
